# Prediction of Wastewater Treatment Plant Performance Using Multivariate Statistical Analysis: A Case Study of a Regional Sewage Treatment Plant in Melaka, Malaysia

Sofiah Rahmat [1,2], Wahid Ali Hamood Altowayti [3,*], Norzila Othman [2,*], Syazwani Mohd Asharuddin [2], Faisal Saeed [4], Shadi Basurra [4], Taiseer Abdalla Elfadil Eisa [5] and Shafinaz Shahir [3]

1. Ministry of Health Malaysia, Kompleks E, Pusat Pentadbiran Kerajaan Persekutuan, Putrajaya 62590, Malaysia
2. Faculty of Civil Engineering and Built Environment, Universiti Tun Hussein Onn Malaysia, Parit Raja 86400, Malaysia
3. Department of Biosciences, Faculty of Science, Universiti Teknologi Malaysia, Johor Bahru 81310, Johor, Malaysia
4. DAAI Research Group, Department of Computing and Data Science, School of Computing and Digital Technology, Birmingham City University, Birmingham B4 7XG, UK
5. Department of Information System-Girls Section, King Khalid University, Mahayil 62529, Saudi Arabia
* Correspondence: ahawahid2@live.utm.my (W.A.H.A.); norzila@uthm.edu.my (N.O.)

**Abstract:** The wastewater quality index (WWQI) is one of the most significant methods of presenting meaningful values that reflect a fundamental characteristic of wastewater. Therefore, this study was performed to develop a prediction approach using WWQI for a regional wastewater treatment plant (WWTP) in Melaka, Malaysia. The regional system of WWTP provides a huge amount of registered data due to the many parameters recorded daily. A multivariate statistical analysis approach was applied to analyze the database. In this approach, principal component analysis (PCA) was used to reduce the dimensionality of datasets obtained from the field municipal WWTP, and multiple linear regression (MLR) was used to predict the performance of WWQI. Seven principal component analyses were derived where the eigenvalue was above 1.0, explaining 71.01% of the variance. A linear relationship was observed ($R^2 = 0.85$), $p$-value < 0.05, and residual values were uniformly distributed above and below the zero baselines. Therefore, the coefficients of the WWQI model are directly dependent on influent biological oxygen demand (BOD), effluent BOD, influent chemical oxygen demand (COD), and effluent COD values. The experimental results showed that the model performed well and can be used to predict WWQI for each WWTP individually and provide better achievements.

**Keywords:** WWTP; WWQI; PCA; MLR; wastewater

## 1. Introduction

Water is particularly important for the survival of living organisms in the world [1]. Human activities, however, including population increases, growing industrialization, and urbanization, have induced rapid pollution, undermining the supply of drinking water [2]. Moreover, the urban system is mainly concerned with water treatment [3]. Wastewater treatment plant monitoring is critical to ensure the effective treatment of wastewater. Despite advances in the design and operation of urban wastewater infrastructure over the last decade, numerous issues related to wastewater's effluent quality still need to be addressed [4]. Daily monitoring of wastewater's physical, biological, and chemical parameters is needed to plan the suitable treatment required to ensure an effective performance process [5]. The sustainable performance of wastewater facilities is based on a consistent operation achieved through energy savings, consumption reduction, and resource recovery to reduce operating costs for handling various types of data monitoring [6]. The selection

method to be used in the treatment system usually depends on the wastewater characteristics. Each treatment has its own constraints, not only in terms of cost but also in relation to feasibility, efficiency, practicability, reliability, environmental impact, sludge production, operation difficulty, pre-treatment requirements, and the formation of chemical residues [7].

To achieve the efficient operation of a wastewater treatment plant (WWTP), complex data monitoring procedures should be put into place to measure a number of parameters such as the biological oxygen demand (BOD), chemical oxygen demand (COD), dissolved oxygen (DO), total suspended solids (TSS), total oil and grease (O&G), ammonia ($NH_3$), nitrate ($NO_3$), and phosphorous (P), while the pH and temperature need to be accurately monitored. The dynamical behavior of sewage treatment plants is due to the nonlinearity and variations in physical properties in terms of environmental conditions, wide variation in flow rate, and various concentrations of influent composition. In the long term, it will cause difficulties in monitoring, analyzing, and controlling the actual situation of WWTP [8]. Due to the lack of instrumentation, control, and automation technologies, the diagnosis of process performances and plant operations are still conducted by human operators. Various process monitoring and control systems have been introduced such as SMAC (smart control of wastewater system) [9], SCADA (supervisory control and data acquisition) [10], LabView software [11], and biosensor monitoring [12].

Currently, most WWTPs are performed manually while relying heavily on the professional knowledge and experience of technical staff, which has become less effective and risky due to the high error rates caused by human factors, especially with the increasing stringent sewage discharge standards. The operational experience and knowledge for each plant differs according to the wastewater characteristic, type of plant, and equipment efficiency. Human operators observe several variables and apply the most associated ones based on their operational experiences to verify the results. This condition takes a long time to reach a certain conclusion, and results can differ according to their experience [13]. In Malaysia, there are 7000 municipal sewage treatment plants, which cover most of the major cities, and the public municipal treatment plants in total were designed to service 26.9 million population equivalents (PE) with a daily maximum volume of 9.2 million $m^3$ per day. Wastewater predictions for the wastewater quality index (WWQI) are used by government bodies to indicate the quality of the wastewater quality effluent ranging from poor to excellent [14]. This is vital as Malaysia's effluent standards allow for the discharging of treated wastewater to the river, reservoir, and well when it meets the specific criteria for BOD, COD, suspended solids (SS), oil and grease (O&G), ammoniacal nitrogen, nitrate, and phosphorous [15].

WWQI is a single numerical value to represent the quality of wastewater without any unit and can rationally express the data and help to evaluate the overall wastewater quality for various uses. The prediction of wastewater quality standards such as the effluent parameter value and WWQI are valuable and can reduce the number of samplings, energy, and cost. Various available methods to assess the performance of wastewater quality have been studied, such as the British Columbia Water Quality Index (BCWQI), the United States National Sanitation Foundation Water Quality Index (USNSWQI), the Weighted Arithmetic Water Quality Index (WAWQI), the Florida Stream Water Quality Index (FSWQI), and the Canadian Council of Ministers of the Environment Water Quality Index (CCMEWQI) [16].

According to the study conducted by Ramya and Vasudevan, the Canadian method is the most trusted and efficient method to calculate WWQI. Ramya and Vasudevan [17] combined the Canadian method and multivariate statistical analysis, such as PCA and correlation, to develop regression models for WWQI prediction in Tamil Nadu, India, and the COD, BOD, and TDS are considered significant independent variables. Similarly, Pirvu et al. [18] also applied a comprehensive method for calculating WWQI using the Canadian method to present the actual efficiency of the sewage treatment process in Valcea Country, Romania. The method measures the scope, frequency, and amplitude of wastewater quality, in which the higher the score, the better the wastewater quality. Khudhair et al. [19] studied WWQI in Iraq using the weighted arithmetic method based on

water indexing systems by selecting the most influential wastewater parameters such as BOD, COD, TSS, and DO.

Another study by Raut et al. [20] calculated the WWQI using a fuzzy-rule-based approach in a municipal sewage treatment plant in India. Sarkheil et al. [21] discussed the various methodologies of WWQI which can be applied to calculating wastewater performance, as well as the concept of the model and its advantages. The authors expressed that the Canadian method has the advantage of categorizing the frequency and extent to which contaminants depart from their respective standards at each monitoring station compared to fuzzy approaches and aggregative weighted wastewater quality indices. The major influences on the quality performance include the BO, COD, TSS, and pH value. Jamshidzadeh and Barzi [22] developed WWQI using four different aggregation functions, i.e., weighted arithmetic mean function, weighted geometric function, and two weighted mix functions, to evaluate the effluent quality of the North sewage treatment plant in Isfahan for agriculture monitoring.

Principally, better management and proper techniques of evaluation systems can be successfully achieved by the development of WWQI for each WWTP. Multivariate analysis techniques present a higher level of explanation of complex data. The PCA analysis determines the variables that contribute most to the effluent quality. This gives important inputs and information for operators and decision makers to modify treatment processes accordingly [23]. Multivariate statistical techniques provide a mechanism to better quantify wastewater quality and treatment processes. Since each plant is independent, and due to the fact that their data are not homogenous, multivariate analysis is the most effective mechanism to measure wastewater quality [24].

In light of the above, this study aims to predict the WWQI performance using multivariate statistical methods. The temporal characteristic of each parameter has been identified. The performance of STP was assessed, and the quality of effluent water was compared with the Malaysia wastewater effluent standards. Then, the statistical interrelationships between different influent and effluent parameters were investigated to develop a WWQI. Multivariate statistical forecasting techniques are based on measured historical data to numerically express the significant analysis.

## 2. Materials and Methods

This study was performed as follows: (1) data collection from Indah Water Konsortium Sdn Bhd, which is the plant operator of the sewage treatment plant; (2) WWQI was developed for effluent standard; (3) PCA was used for data reduction; and (4) MLR was applied as the prediction model of WWQI. The analysis has been conducted using XLSTAT and Statgraphic statistical analysis software.

### 2.1. Site Description

The selected plant is located in Melaka, Malaysia, as shown in Figure 1. This sewage treatment plant is the largest of its kind in the regional plant in Melaka, with an ultimate design capacity of 360,000 PE or 200,000 m$^3$ per day of treated effluent. This plant has been functional since 2017. The plant is a sequence batch reactor (SBR) system consisting of preliminary, secondary, and tertiary treatment systems. Screening and grit removal was designed for preliminary treatment, and the SBR tank process was for secondary treatment. For tertiary treatment, effluents flow to the disinfection process, and the final effluent is discharged to a nearby drain.

### 2.2. Collecting Data

In this study, actual data from 2017 to 2021 were collected and analyzed. The raw data were given by the sewage plant operator, Indah Water Konsortium Sdn Bhd. The BOD, COD, TSS, ammonia, nitrate, pH, temperature, mixed liquor suspended solids (MLSS), and flow rate are among the characteristics examined. At present, the design of domestic sewage treatment plants in Malaysia is only based on comprehensive indicators of pollutants. The

raw data were sorted and modified as needed, with data below detection limits being switched off or displaced as a value to halve the detection limit. Normal distribution tests were carried out using W (Shapiro–Wilk), A2 (Anderson–Darling), and D (Lilliefors test). The XLSTAT software was used to present a vast data collection using univariate statistical analysis. Then, the described statistical analysis of the regression model was compared with Statgraphics Centurion XVI (Statgraphics Technologies, Inc., The Plains, VA, USA).

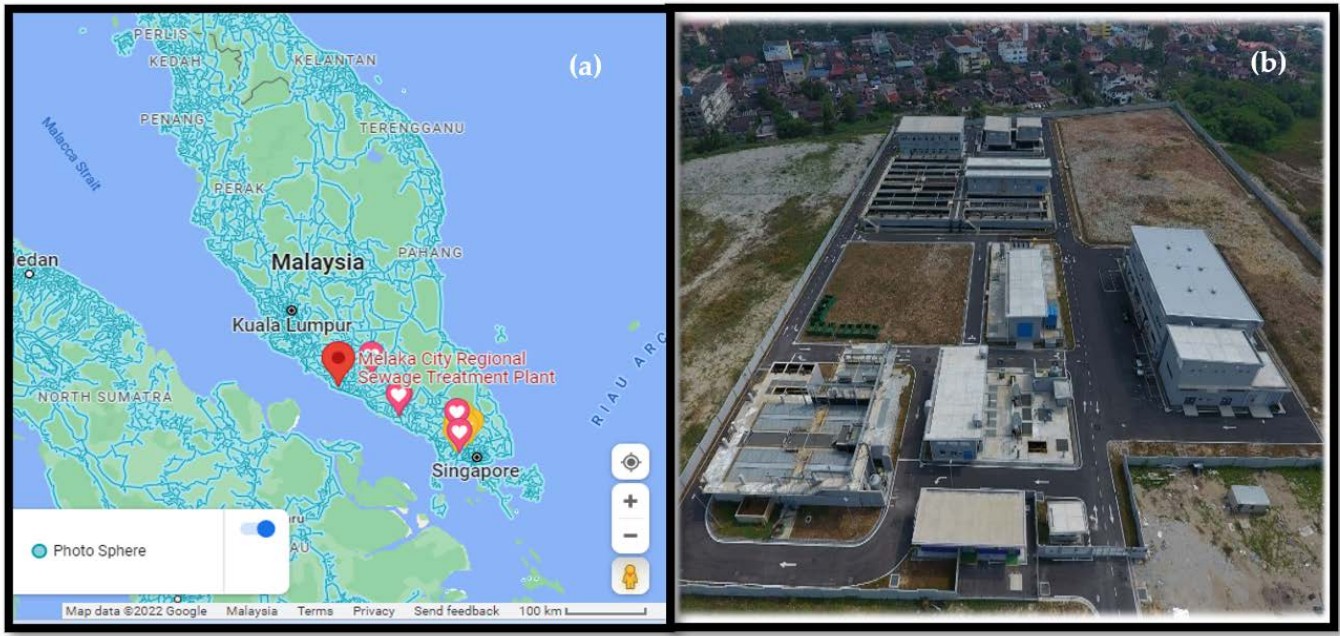

**Figure 1.** (**a**) Site location with 100 km bar distance and (**b**) layout of centralized sewage treatment plant in Melaka, Malaysia.

*2.3. Wastewater Quality Index (WWQI)*

In Malaysia, effluent discharge standards refer to the Environmental Quality Act (Sewage) of 1974, but no specific method was developed for calculating wastewater quality index classification as of 2022. WWQI has been used recently by decision makers to identify the problematic flow of the treatment process or plant. WWQI is an effective mechanism to express the overall condition of wastewater by cumulative consideration of monitored quality indices [25]. The WWQI tries to easily interpret monitored data by ranking the wastewater quality on a rating scale based on measured parameters and established water quality standards. The quality index rating scale is usually from zero to 100. A higher value indicates that wastewater effluents meet the standard set by the regulator, and a higher score represents that the plants are more efficient in operating and monitoring. The WWQI after treatment should have relatively high values, indicating that the effluent is safe and can be discharged directly into water bodies [26].

The WWQI was developed based on the Canadian method [27]. Another study in Baghdad wastewater facilities has applied the same formula from the Canadian wastewater quality index for wastewater evaluation [28]. WWQI is calculated using Equations (1)–(6) as below.

$$F1 = \frac{number\ of\ failed\ parameters}{total\ number\ of\ parameters} \times 100 \tag{1}$$

$$F2 = \frac{number\ of\ failed\ tests}{total\ number\ of\ tests} \times 100 \tag{2}$$

$$Excursion = \frac{failed\ test\ value}{limitation} - 1 \tag{3}$$

$$nes = \frac{\sum_{k=1}^{n} x \ excursion}{number \ of \ tests} \quad (4)$$

$$F3 = \frac{nes}{0.01 \ nes + 0.01} \quad (5)$$

$$WWQI = 100 - \frac{\sqrt{F1^2 + F2^2 + F3^2}}{1.732} \quad (6)$$

where $F1$ is the percentage of measured parameters that do not meet their limit at least once during the period; $F2$ is the percentage of individual tests that do not meet the limitation; $F3$ is the amount by which failed test values do not meet their limitation/objectives; excursion is the number of times by which an individual test is greater than the limitation, and $nes$ is the collective amount by which individual tests are out of compliance. The divisor 1.732 standardizes the resultant quantities of WWQI. The calculation includes three variables, $F1$, $F2$, and $F3$, indicating the percentage of the variables. $F1$ represents the percentage of the variables that quit from the scope objectives, $F2$ represents the percentage of the test frequency that does not match the objective, and $F3$ is counted based on an asymptotic topping amplitude capacity, which scales the standardized entirety of the excursions from the objectives ((nes) (Equation (4)). F3 is evaluated in a three-step process, including the calculation of excursion. Finally, the WWQI is obtained using Equation (6).

### 2.4. Principal Component Analysis (PCA)

The mean, standard deviation, and minimum and maximum values were presented using the data collected in Table 1. The Malaysia Environmental Quality Act (EQA) of 1974 established two criteria for domestic effluent discharge, which are Standard A for discharge upstream of any raw water intake and Standard B for discharge downstream of any raw water intake [29]. From the characterization process conducted, the influent value of BOD, COD, TSS, ammonia, and oil and grease has exceeded the standard A domestic effluent limit by Malaysian Environmental Quality and WHO standards. The mean pH value for both influent and effluent values is 7, indicating that the wastewater is slightly neutral and the same as standard values. It is also found that the concentration of mixed liquor suspended solids (MLSS) is up to 9905 mg/L, indicating biomass nutrient deficiency, bulking sludge, excessive solid generation, high flow rates, and insufficient settling times [30]. The average removal percentage of COD, BOD, TSS, and ammonia was calculated to be 97.24%, 88.92%, 92.55%, and 63.08%, respectively.

There is a complex interrelationship within the wastewater quality variables. The MLSS value fluctuated a lot, while other parameter values were close to each other. It is difficult to utilize one parameter or one set of data to interpret variable variation without a clear understanding of the treatment process. The fundamental concept of PCA is to transform a large set of data containing associated variables into a smaller set of uncorrelated variables while maintaining the largest amount of information relating to the variation between the variables in the original data set [31].

The application of PCA in data analysis for biological wastewater treatment can facilitate the detection of data abnormalities, extraction of useful information from undesired interferences, and access to the composition of wastewater to establish a relationship between quality parameters and sources of pollution. The eigenvectors create the new reference system, providing the maximum converted data resolution after being rearranged according to the appropriate eigenvalue reduction. PCA then ranks the eigenvector matrix according to the decreasing quantity of the corresponding eigenvalue [32].

The extracted variables can be implemented for monitoring subsequent changes in the new variance between variables. The $j$th principal component $l_j$ can be expressed as the linear combination of the measured variables, $x$, and associated weighting factors loading, $v$, as presented in Equations (7)–(10) [33]:

$$l_j = v_j \ x_1 + v_2 \ x_2 + \dots v_{jm} \ x_{1m} \quad (7)$$

The linear variables can be reduced to:

$$l_j = v_j{}^T x \qquad (8)$$

where $v_j{}^T$ is a vector containing all the *j*th loadings, and $l_j$ has the greatest variance subject to two conditions:

$$v_j{}^T v_j = 1 \qquad (9)$$

$$v_j{}^T v_i = 0 \ (i < j) \qquad (10)$$

The Varimax rotation method was applied to clearly interpret the PCA's analytical output. By rotating the factor axis, the loading of PCs was redistributed and polarized, and the resulting new variables were termed varifactors. The dataset was standardized before the analysis in order to guarantee equal weights for all parameters during the computing process. A corresponding factor loading larger than 0.75 is defined as strong, a range of 0.5 to 0.75 is defined as moderate, and a range of 0.3 to 0.5 as weak [34].

**Table 1.** Variables used for PCA.

| Variable | Description | Unit | Minimum | Maximum | Mean | Std. Deviation ($\pm$) | Malaysia Effluent Standard (A) |
|---|---|---|---|---|---|---|---|
| BOD | Influent BOD | mg/L | 6.00 | 350.00 | 133.76 | 66.34 | - |
| $COD_i$ | Influent COD | mg/L | 30.00 | 1323.00 | 307.62 | 213.98 | - |
| $TSS_i$ | Influent TSS | mg/L | 15.00 | 903.00 | 148.48 | 137.08 | - |
| $Ammonia_i$ | Influent ammonia | mg/L | 8.00 | 38.00 | 20.64 | 6.49 | - |
| $pH_i$ | Influent pH | - | 6.40 | 8.20 | 7.00 | 0.30 | - |
| $O\&G_i$ | Influent O&G | mg/L | 1.00 | 135.00 | 32.12 | 24.28 | - |
| $BOD_e$ | Effluent BOD | mg/L | 2.00 | 18.00 | 3.68 | 2.30 | 20 |
| $COD_e$ | Effluent COD | mg/L | 20.00 | 76.00 | 34.07 | 10.64 | 120 |
| $TSS_e$ | Effluent TSS | mg/L | 2.00 | 42.00 | 11.06 | 7.59 | 5 |
| $Ammonia_e$ | Effluent Ammonia | mg/L | 1.00 | 45.00 | 7.62 | 6.75 | 5 |
| $pH_e$ | Effluent pH | - | 6.00 | 8.50 | 7.02 | 0.33 | 6–9 |
| $OG_e$ | Effluent O&G | mg/L | 1.00 | 7.00 | 2.07 | 1.08 | 5 |
| $Temp_e$ | Effluent Temp | °C | 29.00 | 32.00 | 29.55 | 0.59 | 40 |
| $Nitrate_i$ | Influent Nitrate | mg/L | 1.00 | 11.00 | 1.05 | 0.74 | - |
| $Nitrate_e$ | Effluent Nitrate | mg/L | 1.00 | 24.00 | 1.57 | 2.46 | 20 |
| MLSS | MLSS | mg/L | 2918.00 | 9905.00 | 5858.94 | 2226.53 | - |
| WWQI | WWQI | - | 36.30 | 86.80 | 74.30 | 9.10 | - |

### 2.5. Multiple Linear Regression Analysis (MLR)

The MLR method can be applied in a WWTP to predict the effect of two or more independent variables such as BOD, COD, TSS, temperature, pH, and ammonia for WWQI. In addition, MLR models the connection between descriptor variables and a response variable by fitting a linear formula for the observed data. Due to these benefits, the MLR method is used in various environmental studies [35].

Initially, the specified parameters are examined with PCA to check their connection to WWQI. The weakly correlated parameters will be excluded to predict the WWQI. The correlated parameters will be used in the MLR process. The most common measure of correlation is Pearson's correlation, which is commonly referred to simply as the correlation coefficient (r) and is expressed in the following equation [36].

$$r = \left( \sum_{i=1}^n x \frac{(x_i - x)(y_i - y)}{\sqrt{\sum_{i=1}^n x(x_i - x)^2 \sum_{i=1}^n x(y_i - y)}} \right) \qquad (11)$$

where $x_i$ is the real observed value, $y_i$ is the predicted values, n is the number of values, $x$ is the average of observed values, and $y$ is the average of predicted values.

In other terms, the multiple linear regression determines the fitness of a linear relationship between one dependent variable denoted by $\gamma$ and other independent variables denoted by $x_i$. The evolved MLR model can be represented in the following equation:

$$\gamma = \beta_o + \beta_1 x_1 + \beta_2 x_2 + \beta_3 x_3 + \ldots. + \beta_n x_n \tag{12}$$

where $\beta_o$ is the intercept of the regression line, $\beta_1$ is the regression coefficient (slope), $\gamma$ is the dependent variable to be predicted, and $x_i$ is the independent variable [37]. The coefficient of determination ($R^2$), Pearson (R), and standard error are the governing factors for indicating the strength of the model. The strength of the pairwise correlations was evaluated using Pearson's correlation coefficients which have a value between +1 and $-1$, where +1 indicates total positive linear correlation, 0 indicates no linear correlation, and $-1$ indicates total negative linear correlation. A higher $R^2$ indicates that the model has a strong predictive power [38]. The performance of the developed models was assessed by $R^2$ and the root mean squared error (RMSE). The *p*-value will rule the decision by choosing variables that have a significant value ($p < 0.05$). For all calculations, the XLSTAT software (Addinsoft company, Paris, Ile-de-France, France) and Statgraphics Centurion XVI (Statgraphics Technologies, Inc., The Plains, VA, USA) were used. Both models were compared to study the accuracy of their prediction. The comparison was performed on the basis of the relevant statistical metrics such as the correlation coefficient, $R^2$ value, mean square error, and standard error.

## 3. Results

### 3.1. Wastewater Quality Index (WWQI)

The WWQI uses several categories to explain the overall condition of the wastewater. These are presented in Table 2 as the following classes of wastewater quality: 95–100 is an excellent condition; 80–94 is a good condition; 65–79 is a fair condition; 45–64 is a marginal condition; and 0–44 is a poor condition [16].

**Table 2.** Wastewater quality category based on CCME WWQI.

| Quality Range | WWQI | Category |
|---|---|---|
| Excellent | 95–100 | Very close to natural or pristine levels |
| Good | 80–94 | Rarely depart from natural or desirable levels |
| Fair | 65–79 | Sometimes depart from natural or desirable levels |
| Marginal | 45–64 | Often depart from natural or desirable levels |
| Poor | 0–44 | Quality is almost always threatened or impaired |

The WWQI line graph shown in Figure 2 is based on the outgoing wastewater reading of BOD, COD, nitrate, pH, and TSS concentration. These parameters are the main organic and inorganic contaminant indicators that characterize the overall quality of wastewater. These parameters are also the main criteria listed in the effluent standard under Malaysia's Environmental Quality Act (Sewage) of 1974. The WWQI shows a variation in fluctuations. As calculated, the average reading of WWQI was consistently around 75% to 85%, indicating that the effluent quality was in a fair and good index.

The boxplots for the BOD and COD measurements were created to detect series variability and the occurrence of outliers. All wastewater parameters in Figure 3 present outliers. The high values recorded for influent COD and other values were observed to have a low impact on WWQI. The outliers in the box plot arise due to organic shock load or hydraulic flow increase during certain seasons. The influent BOD and COD values do not meet the design standard of municipal wastewater. The average municipal wastewater demonstrates a value of BOD from 200 mg/L to 300 mg/L and a COD value from 300 mg/L to 500 mg/L. This condition of wastewater indicates that too much organic matter was present in the incoming sewage and the illegal discharge of pollutants into the domestic sewer system. Such a situation was observed when influent BOD and COD was 350 mg/L

and 1323 mg/L, respectively. The acceptable levels of BOD and COD for standard A in plant effluent are 20 mg/L and 120 mg/L, respectively. Despite the presence of large deviations from the typical wastewater composition, these scenarios did not affect the STP's operation and maintenance. The composition of domestic wastewater in Malaysia's sewerage system is not significantly different from the composition observed in sewerage systems in other cities in Algeria [39] and Turki [40]. However, the average value of effluent standards for BOD and COD were 18 mg/L and 76 mg/L, respectively, which is below the set maximum limit of Malaysia effluent discharge.

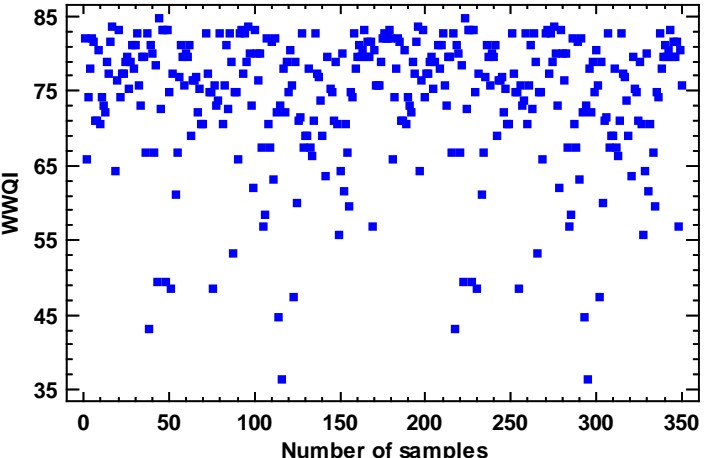

**Figure 2.** Wastewater Quality Index variation.

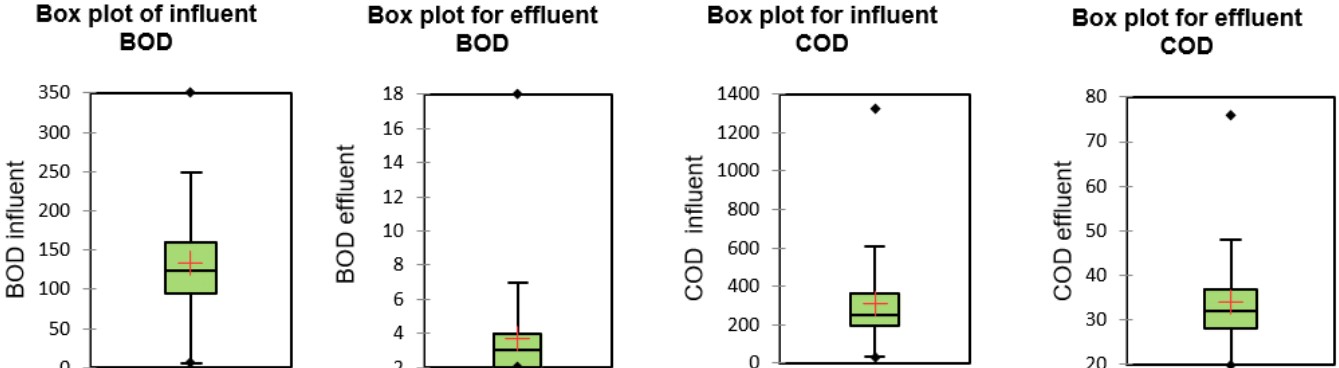

**Figure 3.** Boxplots of the wastewater parameters; from left to right: influent BOD, effluent BOD, influent COD, and effluent COD.

Overall, the performance of WWTP is in good condition since the mean value of parameters meets the standard limit. A review studied by Saidulu et al. [41] concluded that the biological treatment process is preferred as a suitable treatment technique due to its technical simplicity and cost-effectiveness in wastewater management. The typical biological wastewater treatment, such as sequencing batch reactors, produces less sludge and can achieve effluent standard limits without chemical usage [42].

*3.2. Principal Component Analysis (PCA)*

PCA was carried out for all seventeen parameters. The analysis extracted seven principal factors based on Kaiser's rule of eigenvalues more significant than one, as shown in Table 3. Each component explained a specific variance percentage of variables which refer to component loading. The derived features described approximately 71.01% of the total variability of the dataset. The dimension of the dataset was decreased to five factors, as shown in Table S1 (Supplementary Material). The highest loading was explained to

PC1, which accounted for 20.39% of the total dataset's variance, as shown in Figure 4. The first factor (PC1) represented the effluent quality parameters' loading and provided the dominant pattern. The parameter correlated with PC1, such as effluent value for BOD, COD, TSS, and ammonia. The second component (PC2), accounting for 15.94% of the data variance, contained significant loading for the influent quality parameters, such as influent value for BOD, COD, TSS, and oil and grease. The absolute value loadings for all these parameters are greater than 0.3. The third component (PC3), related more to physical characteristics, includes influent pH, effluent temperature, and effluent nitrate.

**Table 3.** Selected principal components with PCA.

|  | F1 | F2 | F3 | F4 | F5 | F6 | F7 |
|---|---|---|---|---|---|---|---|
| Eigenvalue | 3.466 | 2.710 | 1.520 | 1.268 | 1.067 | 1.031 | 1.008 |
| Variability (%) | 20.389 | 15.944 | 8.941 | 7.462 | 6.274 | 6.067 | 5.930 |
| Cumulative (%) | 20.389 | 36.333 | 45.273 | 52.735 | 59.009 | 65.076 | 71.006 |

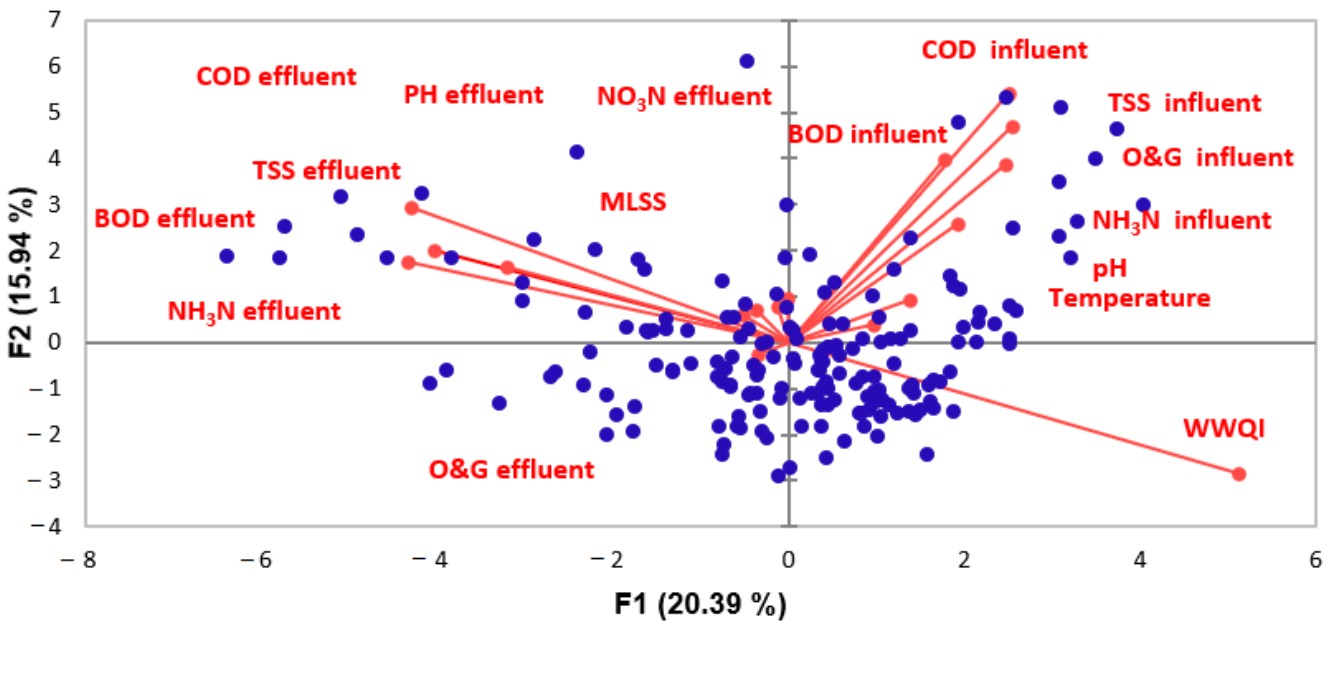

**Figure 4.** The principal component analysis diagram for the active variables and active observations; PCA 1 for effluent parameters and PCA 2 for influent parameters.

While the fourth component (PC4) was mainly related to sludge stability, MLSS, and nitrate, when the sludge concentration increased, the process showed a good level of total nitrogen removal. Among the factors influencing nitrate removal in a sewage treatment process is the concentration of MLSS. A study by Gao et al. [43] confirmed that the nitrate removal rate was positively related to MLSS concentration, with a high correlation coefficient. These results agreed with the fact that the high MLSS in the anoxic zone had a better effect on nitrogen removal. The last component (PC5) accounted for the effluent pH and effluent oil and grease. Traditionally, breaking oil emulsion out via pH adjustment has been the most utilized and effective way to remove oil from wastewater.

For the quality parameters of wastewater tested in the Melaka sewage treatment plant, Person correlation coefficients were determined as presented in Table S2 (Supplementary Material). The results indicate strong correlations between the basic parameters of wastewater quality;

and between influent BOD and influent COD, influent TSS, influent ammonia, and influent oil and grease. Very strong positive Pearson correlations were observed between influent TSS and influent COD, and a strong negative correlation occurs between effluent COD and WWQI. The COD effluent was significantly correlated with WWQI, which showed the highest correlation value (r = −0.906), then followed by TSS effluent (r = −0.791), BOD effluent (r = −0.678), and ammonia effluent (r = −0.454) [44,45].

### 3.3. Multiple Linear Regression (MLR)

3.3.1. XLSTAT Modeling

For the XLSTAT analysis, a regression analysis was used to verify the accuracy of the calculated WWQI. After testing the model's dataset in various scenarios of each PCs, only PC1 and PC2, which are influent index and effluent index variables, respectively, have the best performance in predicting WWQI.

The developed model for predicting the WWQI was based on historical data on the influent index and effluent index of BOD and COD values. In other scenarios, although more parameters were introduced to the models, the accuracy decreased with $R^2$ to less than 0.85. There are several factors that affect the model's performance such as data variation and data quantity. According to Abba et al. [23], for a good analysis of any data intelligence model, the efficiency performance should include a larger value of goodness-of-fit ($R^2$) and lower absolute error measure (e.g., RMSE). Further examination of the models revealed that MLR projected values had a high level of accuracy, as shown in Figure 5.

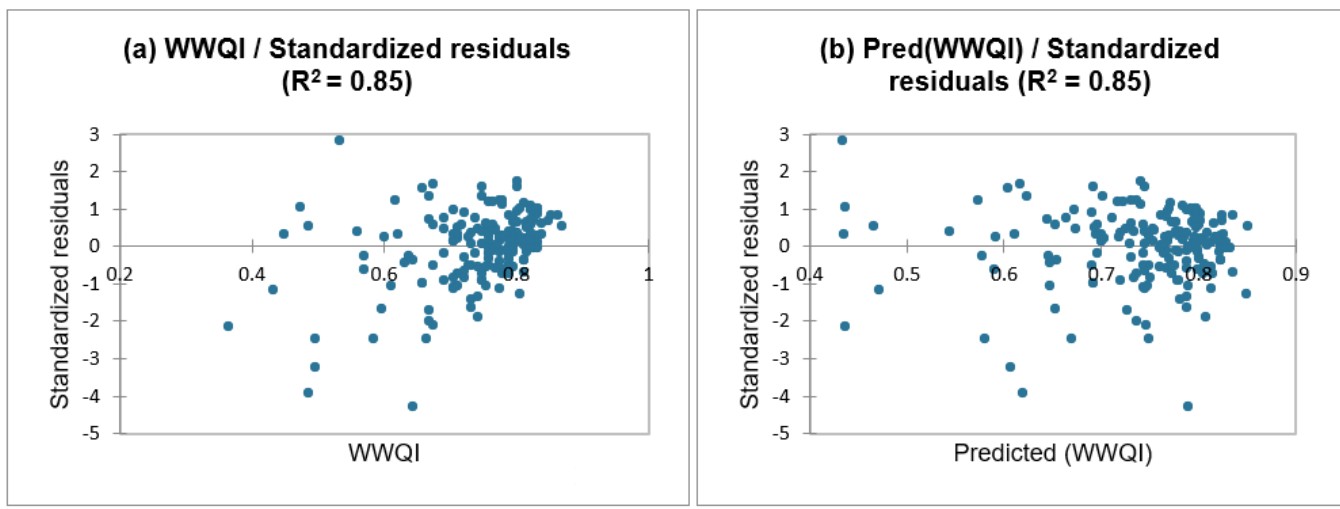

**Figure 5.** The plot of residuals versus (**a**) WWQI and (**b**) Predicted WWQI by XLSTAT model.

Analysis of variance (ANOVA) results of the regression coefficients, the sum of squares, and *p*-values were determined. Multiple linear regression analysis was performed to evaluate the statistically significant variables at a significant level of 0.05. The concentrations of variables are expressed in mg/L. The linear regression equation obtained from the XLSTAT model is as follows:

$$WWQI = 0.993 + 0.000018\ BODi + 0.000018\ CODi - 0.009\ BODe - 0.007\ CODe \quad (13)$$

The numbers of observations and $R^2$ values are summarized in Table 4. A linear relationship was observed ($R^2$ = 0.85, $p < 0.05$), and regression analysis coefficients for the model are shown in Table 5. This model can be used to estimate the value of WWQI and to predict the wastewater treatment plant performance immediately. When $R^2$ is more than 0.85, this shows that the model appears to be confirmed, indicating that this technique is suitable for modeling WWQI [46].

**Table 4.** Statistical evaluation of XLSTAT model summary for the effluent WWQI.

| Source | DF | Sum of Squares | Mean Squares | F | Pr > F | $R^2$ | Adjusted $R^2$ |
|---|---|---|---|---|---|---|---|
| Model | 4 | 2.511 | 0.628 | 530.587 | <0.0001 | 0.856 | 0.856 |
| Error | 353 | 0.418 | 0.001 | | | | |
| Total Corrected | 357 | 2.928 | | | | | |

**Table 5.** Regression coefficients analysis for the XLSTAT model.

| Source | Value | Standard Error | t-Value | *p*-Level |
|---|---|---|---|---|
| Intercept | 0.993 | 0.007 | 138.313 | <0.0001 |
| $BOD_i$ | 0.000018 | 0.000033 | 0.563 | 0.574 |
| $COD_i$ | 0.000018 | 0.000010 | 1.813 | 0.071 |
| $BOD_e$ | −0.009 | 0.001 | −8.972 | <0.0001 |
| $COD_e$ | −0.007 | 0.000 | −31.286 | <0.0001 |

According to the statistical analysis, the residuals of the models follow a normal distribution, and the mean value of the residuals should be zero. Otherwise, it should be suspected that there is a calculation error or the insertion of an additional variable for the regression model should be added. For the proposed WWQI model, the residual values are uniformly distributed above and below the zero baselines, as shown in Figure 6. Therefore, WWQI model coefficients are directly dependent on BOD and COD values [47]. It is also observed that these two variables have more effects on the calculated score and increase the group rating value, while other parameters have less impact on the $R^2$.

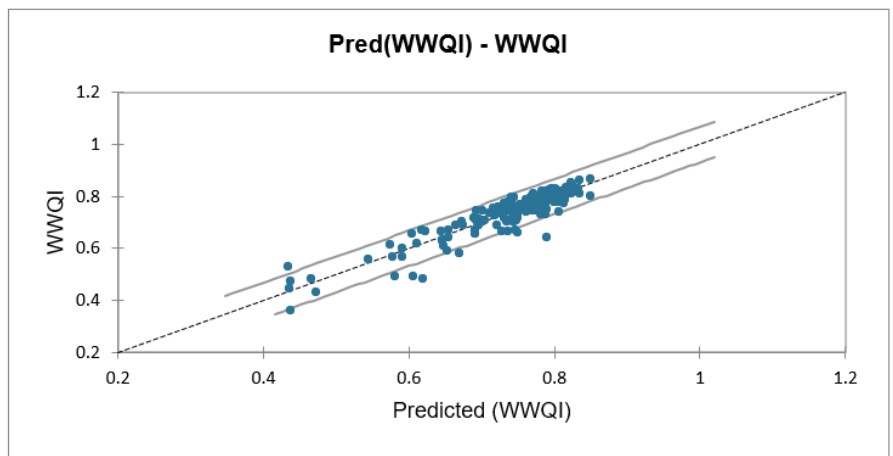

**Figure 6.** Comparison between predicted and observed values for the WWQI by XLSTAT model.

3.3.2. Statgraphic Modeling

The $R^2$ score for the stratigraphic model reveals that the model as fitted explains 85.72% of the WWQI variability, compared to the XLSTAT model, which explains 85.60%. The modified $R^2$ statistic is 85.60 for both models, which are suited for comparing models with varied numbers of independent variables. The standard error of the estimated value for Statgraphic and XLSTAT models is 3.43 and 3.40, respectively.

The mean absolute error (MAE) for the Statgraphic model is 2.46 and 3.64 for the XLSTAT model, which explains the mean of all percentage errors between the predicted and actual value. Since the *p*-value is greater than 0.05 for both models, there is no indication of serial autocorrelation in the residuals at the 95% confidence level. Table 6 shows the statistical evaluation of the model summary, and Table 7 shows the regression analysis for

the Statgraphic model. The linear regression equation obtained from the Statgraphic model is as follows:

$$WWQI = 99.4487 + 0.002145\,CODi - 0.872038\,BODe - 0.66303\,CODe \tag{14}$$

**Table 6.** Statistical evaluation of the Statgraphic model summary for the effluent WWQI.

| Source | DF | Sum of Squares | Mean Squares | F | Pr > F | R² | Adjusted R² |
|---|---|---|---|---|---|---|---|
| Model | 3 | 25,103.2 | 8367.72 | 708.59 | <0.0001 | 85.7245 | 85.6035 |
| Error | 354 | 4180.38 | 11.809 | | | | |
| Total Corrected | 357 | 29,283.5 | | | | | |

**Table 7.** Regression coefficients analysis for the Statgraphic model.

| Source | Value | Standard Error | t-Value | *p*-Level |
|---|---|---|---|---|
| Intercept | 99.4487 | 0.65471 | 151.898 | <0.0001 |
| $COD_i$ | 0.002145 | 0.000854 | 2.5107 | 0.0125 |
| $BOD_e$ | −0.872038 | 0.097284 | −8.96375 | <0.0001 |
| $COD_e$ | −0.66303 | 0.0211011 | −31.4215 | <0.0001 |

The difference between the observed and predicted values by the XLSTAT and Statgraphic models were also compared using the parity plots, as shown in Figure 7. Among the parameters, the influent index and effluent index of BOD and COD had the best matching with the real dataset; some other researchers reported the same finding [48]. The predictions of the regression models were compared with the calculation resulting from actual collected values and showed 85% accuracy. The standard error showed the error associated with the coefficients. The significance of the *p*-value was 0.0001, which indicates 99% significance [49]. The results indicated that BOD and COD strongly impacted WWQI. Our study results show similar trends to the outcomes reported by another study that used a similar methodology as the Canadian method and had applied it to Oran City, Algeria, for assessing WWQI [50]. Aboulfotoh [51] also conducted WWQI in Egypt's WWTP. They utilized multiple linear regression to examine the influence and effluence of the WWQI variables. Having said that, sufficient data must be available, as regression models significantly reduce the amount of data that can be used to maximize the R² value. Still, these findings do not differ from those of other researchers. Sarkheil et al. [21] confirmed that the dominant pollutant in both Fuzzy WWQI and aggregative weighted WWQI methodologies was the BOD parameter, which is 65.38% of the analysis, while the COD parameter is dominant about 34% of the time. Previous studies have supported the application of XLSTAT and Statgraphic in the modeling regression process [52,53].

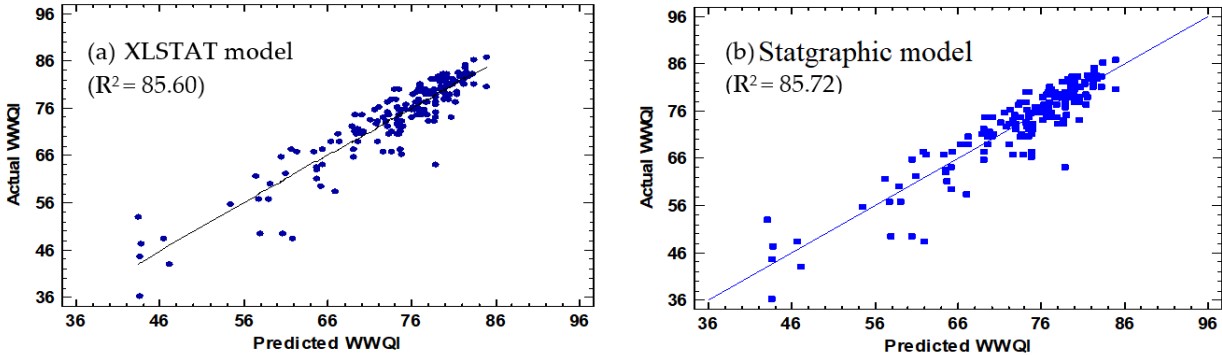

**Figure 7.** Comparison of the model for the measured and predicted value for the WWQI model; (**a**) XLSTAT analysis and (**b**) Statgraphic analysis.



## 4. Conclusions

The purpose of this study was to develop a practical method to predict WWQI which operates on the individual plant to analyze and identify the problem in real time. The data on the process were effectively extracted and modeled by applying a multivariate statistical approach, such as PCA and MLR. The PCA method is one of the most effective methods for understanding large datasets in wastewater treatment experiments. The PCA applications demonstrate good data reduction in an interpretable manner, easily understood and while preserving most of the non-relevant information in the dataset. Interpretable correlations of wastewater quality parameters can be achieved by reducing the data's dimensionality from the original 17 variables to 7 principal components, explaining 71% of the total data variance.

As a result, WWTP could be classified into five PCs according to effluent concentration, influent concentration, physical characteristic, sludge settle ability, and oil removal from the process. Furthermore, our approach has been developed with the aim of minimizing the processing time that is usually required by human operators. The modeling of WWQI by MLR can be considered an expert tool for planners and non-expert staff to monitor and assess wastewater quality daily rather than analyzing all of the parameters. WWQI also offers a better estimation performance for the overall wastewater quality. In the future, more wastewater indicators will be considered during the model construction besides other techniques to enhance the accuracy and the performance effectiveness through investigating and assessing the performance of sewage treatment plant quality in Malaysia.

**Supplementary Materials:** The following supporting information can be downloaded at: https://www.mdpi.com/article/10.3390/w14203297/s1, Table S1: The relationships between the variables were examined by the Pearson correlation matrix. Table S2: Pearson correlation matrix for wastewater quality parameters for all sampling points.

**Author Contributions:** Conceptualization, S.R. and W.A.H.A.; methodology, S.R. and W.A.H.A.; software, S.R., W.A.H.A., F.S., S.B.; validation, S.R., W.A.H.A., F.S., S.B.; formal analysis, W.A.H.A., F.S., S.B., T.A.E.E.; investigation, S.R. and S.M.A.; resources, S.R.; data curation, S.R., W.A.H.A., S.M.A., T.A.E.E.; writing—original draft preparation, W.A.H.A., S.R.; writing—review and editing, W.A.H.A., N.O., S.M.A., F.S., S.B., T.A.E.E.; visualization, S.M.A. and S.S.; supervision, W.A.H.A. and N.O.; project administration, N.O., S.M.A. and S.S.; funding acquisition, T.A.E.E., N.O. and S.S. All authors have read and agreed to the published version of the manuscript.

**Funding:** Deanship of Scientific Research at King Khalid University through Large Groups (Project under grant number (RGP.2/49/43)). Ministry of Higher Education (MOHE) through Prototype Research Grant Scheme (PRGS/2/2020/WAB02/UTHM/02/1). Universiti Teknologi Malaysia (UTM) for financial sponsorship and the Post-Doctoral Fellowship Scheme under the Professional Development Research University Grant (06E27).

**Institutional Review Board Statement:** Not applicable.

**Informed Consent Statement:** Not applicable.

**Data Availability Statement:** Not applicable.

**Acknowledgments:** The authors fully acknowledged the Deanship of Scientific Research at King Khalid University for funding this work through Large Groups (Project under grant number (RGP.2/49/43)). The authors extend their appreciation to the Ministry of Higher Education (MOHE) through Prototype Research Grant Scheme (PRGS/2/2020/WAB02/UTHM/02/1). This study was supported financially by the Ministry of Health Malaysia, and data were provided by Indah Water Konsortium Sdn Bhd, Melaka, Malaysia. Wahid Ali Hamood Altowayti extends his gratitude to Universiti Teknologi Malaysia (UTM) for financial sponsorship and the Post-Doctoral Fellowship Scheme under the Professional Development Research University Grant (06E27).

**Conflicts of Interest:** The authors declare that they have no known competing financial interest or personal relationships that could have appeared to influence the work reported in this paper.

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
