# Peer review of "Prediction of Wastewater Treatment Plant Performance Using Multivariate Statistical Analysis: A Case Study of a Regional Sewage Treatment Plant in Melaka, Malaysia"

_water, doi:10.3390/w14203297_

Round 1
Reviewer 1 Report
General: This manuscript entitled on “Prediction of Wastewater Treatment Plant Performance using Multivariate Statistical Analysis: A Case Study of Regional Sewage Treatment Plant in Melaka, Malaysia” develops a practical method with statistical approach for predicting WWQI of wastewater treatment plant. Authors presented a great effort to prepare this manuscript, however, the reviewer has some queries about this paper. I would strongly suggest to check this manuscript by English experts. I suggest that this manuscript can be acceptable with major revision. Other comments are as follows;
- I would suggest the authors to do an extensive literature review (cited) and improve the introduction section.
- Line 69: The term ‘however’ is not usually used when starting the paragraph.
- Line 83-86 need citation.
- Line 88: “In their study, they used…….”. Whose study did authors mention? Please clarify.
- Line 91: “Similarly, [12] also had applied comprehensive method……”. In this sentence, the citation style should be revised by mentioning …..(last name of first author) et al. [12] when starting the sentence with a reference. Similar in Page 4, Line 28.
- Line 79-99 can be combined in one paragraph.
- Line 113: What and who produced 9263 million liters per day? Please clarify.
- Page 4, Line 8-9: Brief description or citation with appropriate references is needed how they examined all of the parameters mentioned here. If the authors analyzed these parameters themselves, the detail procedure can be mentioned in supplementary information and if they obtained all data from other institutions, they should provide the information.
- Page 5, Line 40: Authors stated F1 in Eq (1) as the percentage of measured parameters that do not meet their limit and multiplied by 1. I have a query about it and if I am not wrong, it should be 100 to obtain the percentage. What is the meaning of 1? Authors need to explain it.
- Page 5, Line 70-72: This sentence should be revised to make it understandable for readers (about Standard A).
- Page 8, Line 198: “The variation”. This sentence is incomplete. Authors should be able to review these type of errors in the manuscript before submission.
- Figures 3 and 4 can be compiled within one Figure and box plot can be represented in a single caption (not in each Figure).
- I suggest to make a Table presenting maximum limit of Malaysia effluent discharge for all parameters that evaluated in this study or can be added a column in Table 1.
- The Figures and Tables are not placed at their appropriate position and need to place where they explained for the first time to make easy for reading the text. Also, all the Figure captions are very short and so should be written in detail to make clearly understandable.
- It is better to present principal component analysis (PCA) data in a Figure.
- As mentioned in Section 3.3 and Equation (linear regression model) obtained for WWQI, it seems that only BOD and COD values are important for predicting WWQI. I wonder what happens if the other factors (parameters) are significantly higher in influent and effluent? Are the remaining parameter values not responsible to change the WWQI index? Please explain about it in detail.

Reviewer 2 Report
Manuscript on Prediction of Wastewater Treatment Plant Performance using Multivariate Statistical Analysis: A Case Study of Regional Sewage Treatment Plant in Melaka, Malaysia is well prepared and analysis using standard methodology and tools. This is useful to the readers
Round 2
Reviewer 1 Report
When reading the responses after of the revision of this manuscript thoroughly, it reflects that the authors have done a great effort. They have addressed most of the queries pointed out by this reviewer. The authors well revised and presented the materials. I believe that the revised manuscript is now suitable for acceptance and recommended for publication. However, I don't feel comfortable with the quality of Figures 2 and 6. Authors did not provide line numbers in revised version of manuscript and thus made difficult to check entire paper thoroughly. The quality of these Figures can be improved (at least both axis values and legends). In addition, the PCA should be represented in one Figure (along with Tables) which will be more attractive for readers.
Reviewer 3 Report
There are still many comments have not been reviewed and authors ignored providing any explantion as follow:
1) Figures 2 and 7 did not presented in a professional way.
2) The minus sign (-) instead of hyphen in for all negative values, especially table 5.
3) The data obtained have not been compared with different software rather than XLSTAT? The authors ignored the comment.
4) All the graphs need to be replotted again in a professional way.
Round 3
Reviewer 3 Report
The manuscript can be accepted in present form
